# Earmarking donations to boost study participation? Evidence from a field experiment

Andreas Raff[1]*, Robert Böhm[2,3,4], Christoph Fuchs[1]

1 Faculty of Business, Economics, and Statistics, University of Vienna, Vienna, Austria, 2 Faculty of Psychology, University of Vienna, Vienna, Austria, 3 Department of Psychology and Copenhagen Center for Social Data Science (SODAS), University of Copenhagen, Copenhagen, Denmark, 4 Department of Banking and Finance, University of Innsbruck, Innsbruck, Austria

* andreas.raff@univie.ac.at

## Abstract

Charitable donations are often the most suitable available way to incentivize study participation, yet their optimal design remains unclear. In a preregistered field experiment, we invited 6,711 psychology faculty at top-200 universities to complete a survey in exchange for a US $5 donation to test whether allowing prospective participants to earmark the donation for a specific purpose increases study participation. Contrary to preregistered hypotheses derived from previous literature, the results showed no significant increase in study participation rates when participants could earmark their donation compared to a random allocation of funds. These findings suggest that while earmarking has been shown to enhance overall donation rates, its effectiveness may not extend to incentivizing study participation.

## Introduction

Increasing study participation rates is crucial for conducting robust and reliable academic research. Researchers have long experimented with techniques to raise response rates. The most prominent levers are advance notification, repeated contact attempts, personalization, and the use of incentives [1–3]. Incentives are usually either personal—monetary payments or gifts—or donation-based, whereby the researcher pledges to contribute to charity on the respondent's behalf. While a substantial literature examines how the timing (pre-paid vs post-paid), size, and framing (cash, voucher, lottery) of personal incentives shape study participation [2,4–6], donation incentives have received far less attention. Most existing work has focused narrowly on whether such incentives outperform or underperform personal incentives in attracting respondents for surveys [5–10], with little systematic investigation into how design features of donation incentives perform against each other (one exception is [11]).

Yet understanding how to implement donation incentives optimally to increase response rates is critical. Even if donation incentives are often less effective than

**Data availability statement:** The study materials, supplementary analyses, and data are publicly available from the Open Science Framework repository (https://osf.io/ewz6v/).

**Funding:** The author(s) received no specific funding for this work.

**Competing interests:** The authors have declared that no competing interests exist.

personal incentives [5,6] they can still be the best option when budgetary, ethical or participant-specific constraints rule out personal incentives. For example, some professions—such as public officials, military personnel or university employees—may not be able to accept personal rewards [12–14]. In studies on sensitive topics, participants may hesitate to provide the personal data required for financial transfers. Most importantly, affluent or time-poor individuals may face opportunity costs that no realistic cash payment within the study budget can offset [15,16]. Under such circumstances, even modest personal incentives are unlikely to attract these participants because they lack strong cognitive valuation [17]. Donation incentives, by contrast, can carry an additional affective valuation stemming from the psychological benefits of helping others that makes even a small donation on the participant's behalf more compelling than an equivalently small personal payment [17]. Thus, there are clear scenarios in which personal incentives are not suitable, and donation incentives provide a viable alternative. While this alone warrants research into their optimal design, from a welfare perspective, optimizing donation incentives is valuable in its own right since, they potentially boost response rates while simultaneously directing funds to prosocial causes, creating a dual social benefit.

One promising design variation of a donation incentive that, to our knowledge, has neither been implemented in the context of incentivizing study participation nor studied for its effectiveness in this setting is earmarking—allowing participants to choose the specific cause their donation supports. The charitable-giving literature demonstrates that offering individuals the option to select a specific cause for their donation (e.g., malaria vaccinations in Africa or building schools in Nepal) reliably increases donation willingness [18–20].

In this paper, we investigate whether the earmarking effect generalizes from donation willingness to the context of study participation. Specifically, we test the impact of earmarking within donation-based incentives by comparing a conventional, non-earmarked donation incentive to two forms of earmarking. First, we implement a standard earmarking condition in which participants can select a specific cause to directly receive their donation. Second, recognizing that charities value both increased contributions and flexibility in fund allocation, we further introduce a novel earmarking condition that we coin "Earmarking with Flexibility." In this condition, participants still select their preferred cause, but are informed that the charity may reallocate funds if the chosen cause's funding target has already been met. This flexible approach addresses operational inefficiencies associated with standard earmarking, which often leads to imbalances with some causes becoming overfunded and others underfunded [21,22]. Our flexible earmarking design thus offers potential for a strong Pareto improvement for researchers and charity organizations—boosting participation rates while preserving charities' ability to allocate resources optimally.

In sum, this study assesses whether granting participants control over donation allocation—through either standard or flexible earmarking—can improve the effectiveness of a donation incentive. Our findings contribute to the literature on study incentivization in general—and donation incentives in particular—by being the first to experimentally test earmarking in this context. They provide new insights into both

the potential and the limitations of earmarking for enhancing participant engagement with donation incentives. Moreover, we contribute to the broader earmarking literature by examining the robustness and generalizability of the effect beyond charitable donations—an essential step in evaluating its external validity [23]. Finally, by developing and testing a novel, flexible earmarking format, we broaden the scope of research on earmarking to consider designs that accommodate the operational constraints faced by charitable organizations.

## Study participation

Incentivizing study participation can be understood through the lens of the Leverage–Saliency Theory of Survey Participation [24], which conceptualizes the decision to participate as a cognitive balancing act. Individuals weigh the perceived costs and benefits of participation on a mental scale. Various survey characteristics—such as the study topic, the perceived burden of participation, trust in the researchers, and incentives—exert differential leverage depending on how much influence they have on the decision to participate. Their salience, or psychological prominence at the moment of decision-making, further determines their effectiveness in tipping the scale. Participation becomes likely when salient factors with negative leverage, like time demands or privacy concerns, are outweighed by those with positive leverage, such as the topic of the study or benefits promised via study incentives. Crucially, these benefits need not be monetary. According to the model of impure altruism [25], individuals can experience intrinsic rewards from helping others—for example, through a charitable donation. If study participation triggers such a donation, then increasing its psychological benefits should, in turn, enhance individuals' willingness to participate.

There is substantial literature which has identified a range of design strategies that boost the psychological benefits of donating—and thereby increase the willingness to help. These strategies include, for example, using emotional language, matching donations, storytelling, providing an identifiable victim, or employing earmarking [18–20,26–30]. Yet, these insights have not been applied to the design of donation incentives. This paper takes a first step by focusing on earmarking—a simple, consistently effective way for increasing donations—to spearhead research into optimizing donation incentives.

## Earmarking

The positive effects of earmarking can be readily explained by Self-Determination Theory (SDT) and its sub theory, Cognitive Evaluation Theory (CET) [31,32]. CET states that intrinsic motivation flourishes when two basic psychological needs—autonomy and competence—are satisfied [31,32]. Contexts that meet these needs promote intrinsic motivation and those that thwart them undermine it. Earmarking arguably meets both needs at once.

Granting meaningful choice is the most direct route to supporting autonomy [33]. A meta-analysis by Patall et al. [33] shows that even letting people decide which reward they will receive for an action—labelled choice-of-reward—reliably boosts intrinsic motivation. Earmarking in a donation incentive fits this category: participants choose the charitable project that will receive the donation, thereby exercising genuine choice over their reward for study participation and thereby satisfying their autonomy need.

Earmarking also addresses the need for competence or felt effectance of one's action [34]. When donors steer funds to a specific, clearly defined project, a decision usually made by the charity shifts to them [19]. This renders their contributions' outcome visible and fosters the subjective experience of making a real difference—what the literature calls perceived impact. Empirical studies show that perceived impact boosts donation intentions [27,35,36] and mediates the positive effect of earmarking on giving [18]. Beyond satisfying competence needs, the positive effect of perceived impact on donation willingness enabled by earmarking may partly derive from the sense of personal causation—the feeling that "I, personally, made this happen"—a mechanism highlighted in impact philanthropy [37] which suggests that donors derive value not only from what was achieved but also from the self-attributed role in making it happen.

A collateral benefit of earmarking is increased transparency because it clarifies how donations are used. This financial transparency has been shown to increase trust in organizations [38], and greater trust, in turn, is positively associated with willingness to give [39].

Taken together, these psychological mechanisms help explain why earmarking increases donations [18–20]. While, in theory, earmarking could raise either the amount donated or the likelihood of donating, empirical evidence suggests that its positive impact on donations stems primarily from the latter. Specifically, only Esterzon et al. observed a positive effect of earmarking on donation amount, whereas Fuchs et al. and Özer et al. did not [18,20]. In contrast, all three studies consistently reported that earmarking positively affects the likelihood of donating, indicating that offering earmarking attracts more donors. Thus, when extending these findings to participation incentives, earmarking may similarly broaden the pool of potential respondents and increase study participation rates.

However, it is important to recognize a potentially fundamental distinction between donations paid with one's own money and externally funded donations that may attenuate the effectiveness of donation incentives in general—and of earmarking in particular: participants' psychological ownership of the funds that are donated [40,41]. In a traditional giving framework, donors spend their own resources. A donation is typically conceptualized as reducing personal wealth, i.e., entailing a financial sacrifice that generates the associated emotional rewards [25]. By contrast, incentive-based donations are supplied externally. Because these funds never genuinely belong to participants, they experience no financial sacrifice and therefore may feel less emotionally invested [42]. Consequently, incentivized donations may yield weaker emotional rewards than self-funded donations in general.

Because perceived control breeds psychological ownership [40], giving respondents a say in where the incentive money goes—through earmarking—may partly offset the ownership deficit built into donation incentives. This control may cue a fleeting sense among participants that the money is, in some meaningful way, "theirs." Critically, while earmarking should still enhance perceived impact (satisfying competence needs), reduced psychological ownership may weaken the felt personal impact among participants—specifically, the sense that "I personally caused this change" [37]. Consequently, where donors value personal causation, some emotional rewards may be lost.

Moreover, while the transparency inherent in earmarking may typically foster trust, the motivational impact of trust is likely to be attenuated in this context. Trust only becomes relevant when some degree of risk is present [43]. Consistent with this, empirical research shows that the influence of trust on decision-making tends to be moderated by perceived risk [44,45], with trust becoming more influential as perceived risk increases. In the case of donation incentives, however, risk is minimal: the monetary stakes are typically small, and participants do not contribute their own funds. Consequently, the trust-enhancing aspect of earmarking via transparency may do little to increase motivation in this low-risk setting.

As a result, reduced psychological ownership may diminish both emotional involvement and felt personal causation, potentially limiting earmarking's effectiveness. Combined with trust's attenuated role in low-risk contexts, earmarking offered in a donation incentive may not replicate the motivational pull observed in self-funded donations where donors spend their own money. Still, earmarking has consistently increased donations across diverse studies and settings. Because it satisfies two basic psychological needs identified by CET—autonomy and competence [31]—its motivational benefits should, in principle, generalize to a range of behavioral outcomes beyond charitable giving such as study participation.

## Methods

To test our hypothesis, we conducted a large-scale field experiment targeting a time-constrained population with high opportunity costs—academic scholars—for whom, as previously outlined, a donation incentive is particularly appropriate. The study used real incentives that were paid out to a designated charity organization. Our experiment employed a one-factorial between-participants design with three conditions: Random condition, Earmarking condition, Earmarking with Flexibility condition (see below). This study was preregistered at https://aspredicted.org/5zj6r.pdf.

## Ethics statement

The study was voluntary, involved minimal risk, and did not collect any personally identifiable information. Given the nature of this research, and in line with institutional guidelines set by the Departmental Review Board of the Department of Occupational, Economic, and Social Psychology at the University of Vienna—where ethical approval is generally voluntary—we did not seek formal ethics approval. Written informed consent was obtained from all participants prior to participation.

## Study procedure and experimental conditions

Invitation emails were sent out on January 15, 2024, followed by a reminder on January 24, 2024. The recruitment period ended on February 13, 2024. In both emails, eligible participants were invited to take part in a study on hiring preferences in the field of psychology. The invitation emails began by introducing the study procedure and informing recipients about the estimated time for completion (10 minutes). Further, we notified all email recipients that if they decided to participate, we would donate $5 on their behalf to the Society for the Improvement of Psychological Science (SIPS; https://improving-psych.org/).

Next, we introduced our manipulations to the participants. We informed all of them that the $5 donation would either be randomly allocated to one of three purposes currently supported by SIPS (Random condition) or that they could choose one of the three purposes (Earmarking and Earmarking with Flexibility conditions). The supported purposes included: (i) supporting preprints in psychology (PsyArXiv), (ii) a diversity travel fund, and (iii) a student/postdoc travel fund. The two earmarking conditions differed only in one additional sentence added to the Earmarking with Flexibility condition, which informed participants that SIPS could change the distribution of donations among purposes if the donation goal for a specific purpose was reached.

After clicking on the survey link, but prior to participation, individuals were informed that no risks were expected from the study, that their data would be completely anonymized, and that any findings would only be reported in aggregate form, accessible exclusively to the research team. They were also advised that they could withdraw from the study at any time without penalty. By clicking "I Agree," participants confirmed that they had read the consent form and consented to take part in the study.

After completing the study, participants were asked to provide information on the focus of their scientific work (qualitative, quantitative, or both), their academic position (professor, associate professor, assistant professor, or other), as well as their age and gender.

## Participants

We invited 6,711 academics employed at psychology departments at universities ranked within the top 200 universities globally to participate in our study. The invitations targeted 171 universities from the top 200 universities (based on Times Higher Education Ranking from 2023) that offer psychology programs. The contact details for the 6,711 individuals were manually collected between May 2023 and January 2024 by searching the psychology departments of the respective universities using the Google search engine and copy-pasting the relevant information on academic faculty members. We randomly assigned $n = 2,237$ participants to each experimental condition. Of all the contacted eligible participants, 406 completed the study ($M_{age} = 46.82$, $SD = 10.6$); 50.2% women; 44.3% Full Professors, 22.9% Associate Professors, 23.4% Assistant Professors, 9.4% other).

## Dependent variables

We calculated the ratios of our two main dependent variables for each experimental condition: *survey begun* and *completion*. First, *survey begun* is defined as the number of people who agreed to participate divided by the number of people in the experimental condition (i.e., those who received the invitation email). This measure reflects the number of participants

who started the survey by providing informed consent on the first survey page, regardless of whether they subsequently completed the study. Second, *completion* is defined as the number of people who completed the survey divided by the number of people in the experimental condition. Additionally, for exploratory purposes, we calculated a third dependent variable representing the ratio of people who merely clicked on the survey (derived by counting the unique respondent IDs in the dataset) per experimental condition in the following denoted by *clicked on survey*.

## Analysis strategy

As preregistered, we combined both earmarking conditions for our main analysis, henceforth referred to as Earmarking Combined conditions. Specifically, we tested for differences in the rates of our dependent variables between the Earmarking Combined conditions and the Random condition using a one-tailed two-proportion *z*-test, expecting higher participation rates in the Earmarking Combined conditions than in the Random condition. Similarly, when comparing the earmarking conditions separately with the Random condition, we performed one-tailed two-proportion *z*-tests. As preregistered, since we had no pre-conceived notions about potential differences between the two earmarking conditions, we conducted two-tailed tests to look for possible differences in either direction. Moreover, for all exploratory analyses, we employed two-tailed tests, as well.

Before conducting the study, we ran a sensitivity analysis to assess the appropriateness of our sample size for examining the hypothesis under consideration. To set a plausible baseline participation rate for our sensitivity analysis, we reviewed studies involving the same population—academic psychologists—who were asked to complete studies on a similar topic. Two studies stood out as particularly relevant: Donnelly et al. [46] and Anderson et al. [47]. Donnelly et al. [46] do not report how many individuals were contacted, so no participation rate can be determined. In contrast, Anderson et al. [47] report a 9% participation rate among psychology and management faculty worldwide. Based on this figure—and our own informal experience with similar, time-pressured academic samples—we used 10% as a rounded, illustrative baseline. Using a one-tailed two-proportion z-test with an alpha level of 0.05 and a power of 80%, the sensitivity analysis indicated that our sample size is sufficient to detect an effect where the participation rate in the Earmarking Combined conditions increases to 12% (relative to an assumed participation rate of 10% in the Control condition).

Furthermore, null results in this paper are accompanied by equivalence tests [48], evaluating the hypothesis that the effect size is greater than or at least as great as our smallest effect of interest, $\Delta = 2\%$. To identify a plausible minimum effect size that would justify the additional administrative effort of implementing earmarking, we reviewed meta-analyses on financial incentives in web- and electronic-based surveys, which reveal that such incentives can increase the odds of participation by 1.39 to 2.43 [49–51]. Assuming a baseline participation rate of 10%, these odds ratios correspond to increases in participation of approximately 4–12 percentage points demonstrating that participation can shift substantially in response to incentive structures. Based on this, we set a 2 percentage-point increase as the minimum effect size that would render earmarking practically worthwhile in our context. From the researcher's perspective, an effect of this magnitude could potentially justify the additional organizational effort required to implement earmarked donations, including establishing a collaboration with a charity, ensuring proper fund allocation, and managing follow-up communication. From the charity's perspective, the expected increase in donations could potentially outweigh the operational inefficiencies and reduced flexibility associated with earmarking. If the equivalence test is statistically significant ($p_{eq} < 0.05$), we conclude that the effect is practically equivalent, indicating that the data are most compatible with no significant effect, since it is likely smaller than $\Delta$.

## Results

### Preregistered analyses

First, as preregistered, we examined whether the treatment groups differed in terms of starting the study by providing consent. The results revealed no significant difference between the 7.2% (162 out of 2,237) of the participants in the Random

condition and the 7.5% (336 out of 4,474) of the participants in the Earmarking Combined conditions who gave their consent and began to participate in the study ($z=0.40$, $p=.345$; $p_{eq}=.005$). Attrition among those who began the survey was low and balanced across conditions—17.3% (28/162) in Random, 18.2% (32/176) in Earmarking, and 20.0% (32/160) in Earmarking with Flexibility (all exploratory between-condition comparisons: two-tailed p > .50).

Next, we analyzed whether participants in the different experimental conditions differed in terms of completing the survey. The results revealed that 6.0% (134 out of 2,237) of participants in the Random condition and 6.1% (272 out of 4,474) of participants in the Earmarking Combined conditions completed the study ($z=0.14$, $p=.444$, $p_{eq}=.001$). Moreover, there was no indication that the results differed between the first and the second wave of invitation e-mails (see S1 Table Robustness checks).

As further preregistered, we investigated whether there are differences between the three experimental conditions without conflating the two earmarking groups. As displayed in Table 1, and further underscored by the visual representation in Fig 1, there were no significant differences in any of the outcome measures across all the possible treatment comparisons.

## Exploratory analyses

Regarding the secondary outcome measure of whether participants clicked on the survey link at all, the results revealed that 13.5% (302 out of 2,237) of the participants in the Random condition and 15% (672 out of 4,474) in the Earmarking Combined conditions (Earmarking: 15.7%; Earmarking with Flexibility: 14.3%) clicked on the survey, which is not a significant difference using a two-tailed test ($z=1.6$, $p=.095$, $p_{eq}=.297$).

As shown in S1 Table in the online supplement, participant characteristics across experimental conditions were largely equal regarding, age, whether the participant works mainly quantitatively or qualitatively, ranking of the university, position held, and commitment to open science practices (COSP). Thus, we found no indication that participant characteristics moderate the treatment effects.

**Table 1. Results of significance and equivalence tests.**

| Comparison | Dependent Variables | Differences |
|---|---|---|
| **Preregistered Analyses** | | |
| Earmarking Combined vs. Random | *Survey begun* | **7.5% vs. 7.2%** ($p=.345$; $p_{eq}=.005$) |
| Earmarking Combined vs. Random | *Completion* | **6.1% vs. 6%** ($p=.444$, $p_{eq}=.001$) |
| Earmarking vs. Random | *Survey begun* | **7.9% vs. 7.2%** ($p=.215$, $p_{eq}=.041$) |
| Earmarking vs. Random | *Completion* | **6.4% vs. 6%** ($p=.268$, $p_{eq}=.016$) |
| Earmarking with Flexibility vs. Random | *Survey begun* | **7.2% vs. 7.2%** ($p=.452$, $p_{eq}=.007$) |
| Earmarking with Flexibility vs. Random | *Completion* | **5.7% vs. 6%** ($p=.352$, $p_{eq}=.007$) |
| Earmarking vs. Earmarking with Flexibility | *Survey begun* | **7.9% vs. 7.2%** ($p=.363$, $p_{eq}=.052$) |
| Earmarking vs. Earmarking with Flexibility | *Completion* | **6.4% vs. 5.7%** ($p=.318$, $p_{eq}=.036$) |
| **Exploratory Analysis** | | |
| Earmarking Combined vs. Random | *Clicked on survey* | **15.0% vs. 13.5%** ($p=.095$, $p_{eq}=.297$) |

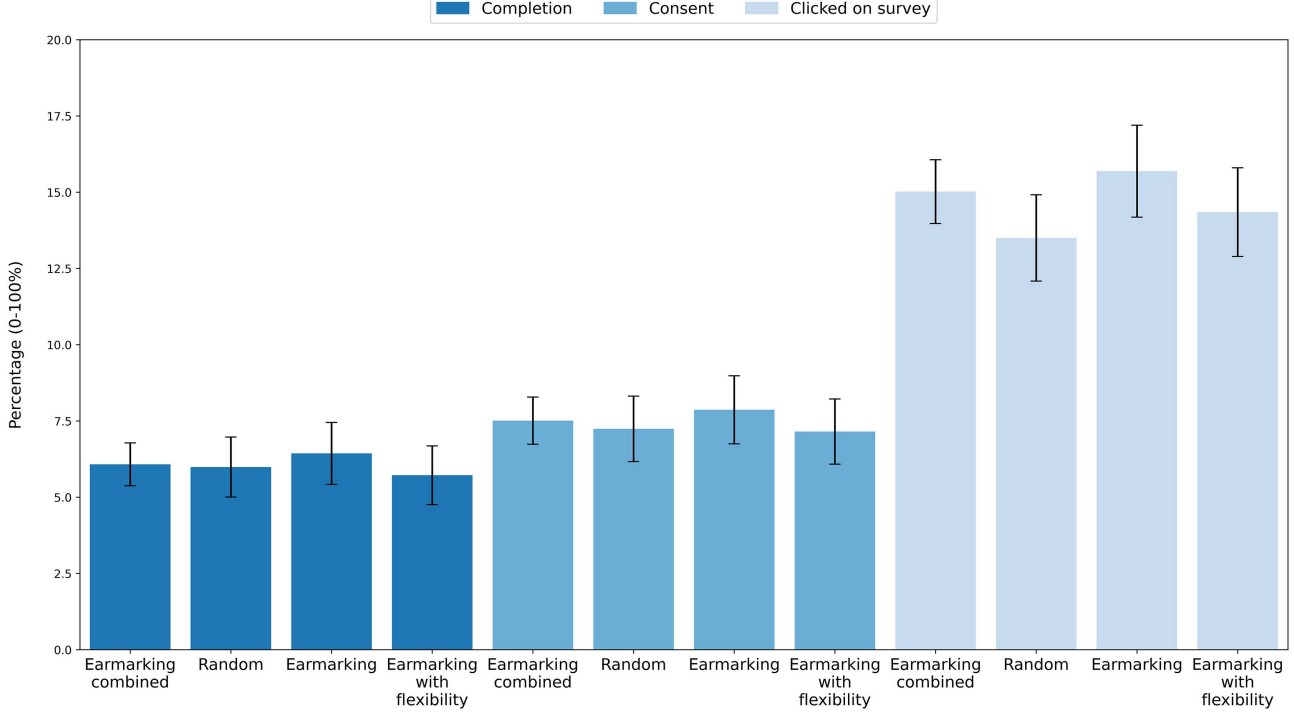

**Fig 1. Results of the three outcome measures.** Proportion of participants who clicked on survey, started the survey, and completed the experiment across experimental conditions, with 95% confidence intervals.

## Discussion

Our results suggest that earmarking did not increase study participation. This lack of evidence of a meaningful earmarking effect was irrespective of the type of earmarking condition and the outcome measure.

### Implications

While future research is needed to better understand and contextualize the absence of effects observed in this study, practical implications can already be drawn from our findings. Specifically, our research shows that earmarking is not an effective way to increase study participation, but its use does not harm participation either. Therefore, earmarking could still be a viable option in contexts where donation-based incentives are deemed appropriate. However, when thinking about implementing earmarking, researchers should weigh its potential benefits against its downsides. As noted earlier, earmarking can reduce organizational flexibility of the recipient charities [21,22], and can impair funding efficiency, project outcomes, and overall impact [52–54]. Moreover, earmarking creates additional administrative burdens for researchers, including coordinating with charities, ensuring correct fund allocation, and managing follow-up communications. Given these costs and the null effects we observed, earmarking should only be integrated into donation incentives when its potential advantages clearly outweigh these downsides.

### Limitations

Despite the strengths of our large-scale field experiment, several limitations emerge from our design—each of which can be meaningfully linked to Leverage–Saliency Theory of Survey Participation.

First, the modest donation amount of $5 may have limited the incentive's perceived value (i.e., leverage), thereby weakening its ability to motivate participation. Although significant earmarking effects have been observed even with very small

sums (as low as $1 [20]), and the emotional rewards of giving are detectable at amounts as modest as $5 (e.g., [55]), future research should explore whether larger donation incentive amounts combined with earmarking enhance leverage and increase participation rates. This may be particularly relevant because, as theorized, psychological ownership of the funds may be lower for donation incentives than for donations paid with one's own money, which may suggest that higher amounts are needed for the positive effects of earmarking to manifest in this context.

Second, our sample—academic scholars—represents a highly distinct population. This group regularly designs and engages with studies and experiments, which might lead to different reactions to experimental stimuli. Extensive experience with experimental paradigms has been demonstrated to potentially attenuate the impact of experimental manipulations [56,57]. According to the Leverage–Saliency Theory of Survey Participation framework, experience may therefore reduce the salience of the incentive and its design variations, thereby diminishing their potential to influence participation. Future studies should examine whether less experienced or more diverse populations respond differently, providing more generalizable insights into earmarking effects.

Third, we did not directly assess participants' trust in SIPS or the importance they attached to its mission. While we believe many participants likely had strong confidence in SIPS's stewardship—given its sustained engagement in the academic-psychology community, commitment to support and promote open and transparent research practices, and annual conferences that draw 500–1,000 attendees—individual perceptions of trust may still meaningfully vary. Likewise, although we chose SIPS to maximize mission importance for participants by aligning the cause with participants' professional identity and values, factors known to increase donation willingness [58,59] we cannot be sure that every participant regarded the mission of the organization as important. Within the Leverage–Saliency Theory of Survey Participation framework, trust in the charity organization and perceived mission importance are potentially important for a donation incentive's leverage: only when the charity is trusted and its mission deemed important variations in the donation incentive (e.g., earmarked vs. non-earmarked) might meaningfully influence participation. If trust or mission importance is low, the incentive's leverage is likely reduced and if trust and mission importance vary widely across individuals, the effect of manipulating donation-incentive design features may be diluted, making true differences harder to detect. Future work should therefore measure trust and mission importance directly to clarify how these attributes potentially moderate earmarking effects.

Fourth, related to the previous limitation, the different purposes to choose from for donations on SIPS (i.e., a preprinting platform and two types of travel funds) are arguably closely related and participants may have felt smaller differences between these purposes than, for instance, different projects supported by the Red Cross, due to its much broader mission and more diverse projects. This limited thematic differentiation may have weakened the perceived meaningfulness of the choice, potentially reducing the leverage of the earmarking manipulation. Because our design therefore minimizes cause differentiation, the earmarking effect we (fail to) detect can be interpreted as a conservative lower-bound estimate. Future research should explicitly vary cause differentiation and test whether thematic distance between causes moderates the earmarking effect.

Finally, a core strength of earmarking lies in its ease of implementation—it can be conveyed in just a few words. However, this simplicity can also be a weakness: the manipulation may be too subtle, leading participants to overlook or fail to register it entirely. Although we attempted to enhance visibility (i.e., saliency) by formatting the manipulation in bold, it remains possible that some participants did not attend to it. Future studies should therefore include manipulation checks to distinguish between those who noticed and understood the experimental manipulations and those who did not.

Considering these limitations, we cannot claim that earmarking is equally inefficient in other contexts and with other samples. We hope that our findings will inspire future research on the generalizability of earmarking effects to different contexts and its potential moderator variables, which may provide important theoretical insights on the very nature of earmarking effects, as well as when they can be expected to be effective or not.

## Future research

A promising direction for future studies is to examine the role of psychological ownership in differentiating donations made with one's own money from externally funded donation incentives, as theorized before, which could be one reason why

earmarking in this particular instance did not prove effective. To investigate this, researchers first need to identify methods for experimentally varying psychological ownership of the incentive funds. The study by Stoffel et al. [60] provides a useful template for achieving this goal. In their research, survey participation increased when respondents had the option to choose between a personal incentive (a $2 Amazon voucher) and a "decoy" donation incentive, compared to a condition that offered only a personal incentive. By providing respondents with a choice to select between the personal and donation incentive, the authors effectively give the participants complete freedom over the funds and therefore maximize respondents' psychological ownership. Future research could investigate this design in combination with earmarking. Although Stoffel et al. [60] found increased survey participation when a donation incentive (for a pre-specified charity organization) was available, ultimately 95% of respondents still chose the personal incentive. Integrating earmarking into this framework could potentially improve social welfare outcomes by not only enhancing participation rates but also encouraging more respondents to opt for donations over personal rewards.

## Conclusion

In summary, our findings suggest that earmarking may not prove effective in motivating study participation. Generalization and extension studies like ours play a critical role in advancing scientific knowledge by highlighting the boundaries and limitations of previously established effects, particularly when extending them to new application domains. They may thus help prevent the overgeneralization of certain effects, ensuring that interventions and strategies are applied appropriately and effectively [61,62].

## Supporting information

**S1 Table. Characteristics of participants.** Participants who completed the study ($n = 406$) dependent to the experimental condition.
(DOCX)

## Acknowledgments

We gratefully acknowledge the cooperation with the Society for the Improvement of Psychological Science (SIPS) in the context of this project. The study generated a total donation of $2,030 to the organization.

## Author contributions

**Conceptualization:** Andreas Raff, Robert Böhm, Christoph Fuchs.

**Data curation:** Andreas Raff.

**Formal analysis:** Andreas Raff.

**Investigation:** Andreas Raff.

**Methodology:** Andreas Raff, Robert Böhm, Christoph Fuchs.

**Visualization:** Andreas Raff.

**Writing – original draft:** Andreas Raff.

**Writing – review & editing:** Robert Böhm, Christoph Fuchs.

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
