## [Decision Letter · Decision Letter 0]

28 Mar 2025

PONE-D-25-07209Earmarking Donations to Boost Study Participation? Evidence from A Field ExperimentPLOS ONE

Dear Dr. Raff,

Thank you for submitting your manuscript to PLOS ONE. After careful consideration, we feel that it has merit but does not fully meet PLOS ONE’s publication criteria as it currently stands. Therefore, we invite you to submit a revised version of the manuscript that addresses the points raised during the review process.

As you can see, both reviewers find the paper interesting but also raise serious concerns about the framing and the study setup.  My own independent reading confirms their views.

As concerns framing, it is unclear how the outcome of "study participation" contributes to the literature on earmarking, specifically its effect on the likelihood of donation. My recommendation is to frame the paper differently, starting with the puzzle of why participation in research studies remains low, and whether the promise of (earmarked) donations can boost participation. This would strike me as a more effective framing strategy and require reviewing a slightly different literature. 

As concerns the study design, it is unclear 1) whether academics are a relevant sample from which we can generalize; 2) whether some outcomes used are meaningful (especially "consent" -- perhaps better labelled as "survey begun"), and 3) if the treatment worked (manipulation checks and power calculations missing). Please add relevant explanations. Moreover, it appears that the results need to be interpreted differently in that any donations for study participation, not just earmarked donations, fail to be effective. This seems like the more important finding but this baseline effect is not discussed. 

We look forward to receiving your revised manuscript.

Kind regards,

Bernhard Reinsberg, Ph.D

Academic Editor

PLOS ONE

Journal Requirements:

2. You indicated that ethical approval was not necessary for your study. We understand that the framework for ethical oversight requirements for studies of this type may differ depending on the setting and we would appreciate some further clarification regarding your research. Could you please provide further details on why your study is exempt from the need for approval and confirmation from your institutional review board or research ethics committee (e.g., in the form of a letter or email correspondence) that ethics review was not necessary for this study? Please include a copy of the correspondence as an ""Other"" file.

4. Please remove all personal information, ensure that the data shared are in accordance with participant consent, and re-upload a fully anonymized data set.

Reviewers' comments:

Reviewer's Responses to Questions

**Comments to the Author**

1. Is the manuscript technically sound, and do the data support the conclusions?

Reviewer #1: Partly

Reviewer #2: Yes

2. Has the statistical analysis been performed appropriately and rigorously? 

Reviewer #1: Yes

Reviewer #2: No

3. Have the authors made all data underlying the findings in their manuscript fully available?

Reviewer #1: No

Reviewer #2: Yes

4. Is the manuscript presented in an intelligible fashion and written in standard English?

Reviewer #1: Yes

Reviewer #2: Yes

5. Review Comments to the Author

Reviewer #1: This article looks at the question of whether the opportunity to earmark donations affects study participation among academics at psychology departments. The motivation for this question is based on the finding in the literature that earmarking donations often affects willingness to donate, and the authors are interested in whether this extends to study participation. The manuscript is clearly written, and the experiment is cleanly executed, which make for an interesting read. However, my main comments are to do with whether study participation is a relevant outcome for donations being earmarked versus not, and whether an academic population is necessarily informative for donation behaviour more broadly. The following points expand on this and summarize a few other comments that I hope will help the authors.

• First, it is not clear why the effect of earmarking donations on study participation is an interesting question because earmarking usually relates to donations to charities and international organizations, as the authors also note, whereas study participation of the sort we see in this manuscript is academic. Therefore, it is not clear why this is an interesting and relevant question and whether (academic) study participation is a relevant and useful outcome for studying the effects of earmarking donations.

• Somewhat related to the previous point, are academics generally a good population for studying the effects of earmarking donations? In other words, academics are not usually the main target audience for charitable donation so studying their receptiveness to earmarking versus not is not necessarily informative. I realize the charity in question is relevant to academics but the literature that the manuscript is situated in is broader and the examples of charities given are also very different from the kind of society/charity used in the experiment. The conclusions drawn seem to be very broad, which is why it becomes even more important to consider whether the type of sample used in the experiment is relevant for the kinds of charities the results are alluding to.

• The choice of the charity makes sense given the target audience, but it is unclear whether it is one that participants would care deeply about especially in comparison to charities that are usually studied in this literature. Did the authors perhaps ask questions to gauge whether participants trusted the Society to choose the right causes, spend the money fully et cetera? Similarly, was this a charity where participants likely cared enough about it in general to care about how their donation money would be spent?

• One of my main concerns when reading about the treatment conditions was whether the purposes were sufficiently different for participants to care enough to want to complete the study and be able to earmark. The authors briefly touch upon this in the conclusion, but this may be a more significant factor than presented because, in the real world, charities, especially international ones, are donating to very disparate causes with a lot of differentiation. For instance, when donating to something like UNICEF, you can often choose between different causes and different countries and, therefore, it is much more likely that those who are donating will care about which cause their donation goes to. In comparison, roughly the same set of recipients between three purposes that are all to do with helping academics within the field of psychology is unlikely to evoke the same reaction or interest in differentiating between the various causes. It may also, in general, not be a set of causes that participants have strong feelings about in any case, again especially in comparison with the types of recipients that large charities, whether national or international, help.

• In the analysis, it is unclear why "consent" is a dependent variable of interest. Participants who consented may or may not have reached the treatment page so it’s not obvious what information can be gleaned from this, and therefore it’s a bit odd that this is a main dependent variable in the analysis.

• The authors summarize their power calculation, but it would be helpful to know whether a change of 2 percentage points would be meaningful in this context, as it seems rather low. It would also be helpful to know what the basis is for the 10% assumed participation rate, unless that is simply an example in which case stating that would add clarity.

Reviewer #2: Introduction

The introduction advances the earmarking concept in charitable giving and provokes the primary question of the study: whether earmarking would boost participation in research studies with donation-based incentives. It defines the gap in generalizing donation behavior findings to study participation and introduces the experimental design.

• Sudden shift to the "dark side" of earmarking. The transition can be facilitated by a bridging sentence that logically links the good and bad sides of earmarking.

• Limited theoretical foundation connecting donation motivation and study participation. Draw on self-determination theory or prosocial behavior spillover literature to account for why earmarking effects would generalize to participation.

Literature Review

Literature review documents prior earmarking studies, psychological accounts (control, effect, transparency) and effect of earmarking on donation behavior.

Surface-level analysis of mechanisms (impact, control, transparency). Expand this section to distinguish between these mechanisms in depth and examine which might not be effective in a study participation context.

Failure to report charity-based rewards in survey research and theories of research participation motivation. Integrate literature on incentives of donation in research and motivational aspects in responding to surveys.

The critique lacks a strong concluding sentence that summarizes the gap and clearly conveys the need for this research.

Methodology

The study uses a large-scale field experiment with 6,711 academic researchers randomly assigned to three conditions: Random, Earmarking, and Earmarking with Flexibility. The main dependent measures are completion and consent rates.

• Uncertainty regarding randomization procedure (manual or computerized, stratified by variables like university?). Detail randomization procedure and include balance checks on age, gender, rank.

• No manipulation check for whether or not participants appreciated and saw the earmarking opportunity. Future experiments must include a post-task assessment of manipulation.

• Equivalence in donation targets erodes the salience of the choice. Future experiments must employ more dissimilar donation opportunities.

• No attrition analysis (who withdrew after consent). Report the drop-off rate and assess if attrition by condition differed.

• Findings show no significant material effect of earmarking on completion or consent rates. Equivalence testing suggests any effect is smaller than the pre-specified 2% cut-off. Click rate exploratory analysis suggests a small, non-significant difference.

Discussion

The discussion interprets null findings, identifies limitations, and specifies implications for the use of earmarking in research volunteer incentives.

• Disjointed limitations section; no flow and connections to theory. Reorganize into a single paragraph with each limitation connected to its effect on results and proposing future research directions.

• Limited discussion of the theoretical explanations for why earmarking may not prompt participation. Incorporate behavioral economic theories or motivation theories describing the distinction between giving and participating.

• The shift to applied implications is abrupt. Add a bridging sentence connecting the theoretical findings to practical recommendations.

6. PLOS authors have the option to publish the peer review history of their article (what does this mean? ). If published, this will include your full peer review and any attached files.

**Do you want your identity to be public for this peer review?** For information about this choice, including consent withdrawal, please see our Privacy Policy .

Reviewer #1: No

Reviewer #2: **Yes: ** Stefanos Balaskas

---

## [Author Response · Author response to Decision Letter 1]

27 Jun 2025

Response to the Editor

Thank you very much for your detailed and constructive feedback on our manuscript, and for recognizing its merit and relevance. Guided by your comments and those of the reviewers, we have undertaken a thorough revision. Below we (i) summarize the most important changes and (ii) respond point-by-point to the specific issues you raised (original comments are italicized).

As concerns framing, it is unclear how the outcome of "study participation" contributes to the literature on earmarking, specifically its effect on the likelihood of donation. My recommendation is to frame the paper differently, starting with the puzzle of why participation in research studies remains low, and whether the promise of (earmarked) donations can boost participation. This would strike me as a more effective framing strategy and require reviewing a slightly different literature.

Following your suggestion, the introduction now opens with the challenge of raising survey response rates and reviews the main strategies scholars currently use—most prominently: incentives. We highlight the surprising under-exploration of how to optimize donation incentives, even though they may be preferable to personal incentives in certain instances. We therefore position earmarking as a promising means of making donation incentives more effective and outline why this possibility has practical and theoretical importance. We hope you will agree that this revised framing now presents a clearer and more compelling account of our study’s contribution.

As concerns the study design, it is unclear 1) whether academics are a relevant sample from which we can generalize

As noted in our response to Reviewer 1, we completely agree that academics might be a population that is typically not the primary target of charitable campaigns. However, we equally see no theoretical reason to assume that academics differ systematically from the general public on traits that drive prosocial behavior. In fact, academics fit our study well precisely because— as the revised manuscript now spells out—they are relatively affluent, highly time-constrained, and, therefore, difficult to motivate with small personal incentives. Accordingly, testing donation‐based incentives in such a population should provide a meaningful test of their effectiveness.

2) whether some outcomes used are meaningful (especially "consent" -- perhaps better labelled as "survey begun")

Following your suggestion and the comment of Reviewer 1, we have now re-labeled the variable formerly called “consent” to “survey begun” throughout the manuscript, thereby clarifying that the variable captures the point at which a participant clicked the survey link and started the questionnaire by providing consent. All text, tables, and figures have been updated accordingly.

and 3) if the treatment worked (manipulation checks and power calculations missing)

We agree that we cannot be entirely sure that the manipulation worked as intended, a point that was also raised by Reviewer 2. Because the email invitation was the only vehicle for the treatment, a post-survey manipulation check would have strengthened internal validity. We explicitly acknowledge this limitation and recommend its inclusion in future work.

Sensitivity analysis: Because the sampling frame comprised all 6,711 eligible academics, N was fixed ex-ante. Instead of an a-priori power analysis, we, therefore, report a sensitivity analysis showing that, at α = .05 and 1–β = .80, the design could detect a ≥ 2-percentage-point lift (12 % vs. an illustrative 10 %).

Moreover, it appears that the results need to be interpreted differently in that any donations for study participation, not just earmarked donations, fail to be effective. This seems like the more important finding but this baseline effect is not discussed.

While we share your intuition given the overall low response rate, we believe that our study design does not allow for a definitive conclusion on this point, as all conditions included a $5 donation incentive. A proper test of that hypothesis would arguably require a true no-incentive control group. In the present manuscript, we, therefore, limit our conclusions to the more specific question of whether adding earmarking to a donation incentive improves participation. Our data indicate that it does not.

We again thank you for your constructive comments and guidance. We believe that our revisions have substantially strengthened the manuscript. Naturally, we are happy to make any further improvements you deem helpful.

Response to Reviewer 1

Reviewer #1: This article looks at the question of whether the opportunity to earmark donations affects study participation among academics at psychology departments. The motivation for this question is based on the finding in the literature that earmarking donations often affects willingness to donate, and the authors are interested in whether this extends to study participation. The manuscript is clearly written, and the experiment is cleanly executed, which make for an interesting read. However, my main comments are to do with whether study participation is a relevant outcome for donations being earmarked versus not, and whether an academic population is necessarily informative for donation behaviour more broadly. The following points expand on this and summarize a few other comments that I hope will help the authors.

Thank you for the overall positive evaluation and the helpful suggestions that helped us to improve the manuscript.

• First, it is not clear why the effect of earmarking donations on study participation is an interesting question because earmarking usually relates to donations to charities and international organizations, as the authors also note, whereas study participation of the sort we see in this manuscript is academic. Therefore, it is not clear why this is an interesting and relevant question and whether (academic) study participation is a relevant and useful outcome for studying the effects of earmarking donations.

Thank you for raising this important point. In the revised manuscript, we now more clearly articulate why examining the effect of earmarking—as a specific way of designing donation incentives—on study participation is both relevant and meaningful. Specifically, we explain that donation incentives may be particularly useful in contexts involving highly affluent, time-constrained populations—such as academics—who are less responsive to small personal incentives due to high opportunity costs. In such cases, donation-based incentives offer a viable and potentially more appropriate alternative. We selected academics as our sample precisely because they exemplify such a time-poor population with high opportunity costs for whom donation incentives are particularly appropriate.

Moreover, despite their practical relevance, there is limited systematic evidence on how donation incentives can be optimized. Examining whether a well-documented design feature to increase donations—earmarking—can increase study participation thus addresses a practical need and extends the theoretical framework surrounding the earmarking construct.

• Somewhat related to the previous point, are academics generally a good population for studying the effects of earmarking donations? In other words, academics are not usually the main target audience for charitable donation so studying their receptiveness to earmarking versus not is not necessarily informative. I realize the charity in question is relevant to academics but the literature that the manuscript is situated in is broader and the examples of charities given are also very different from the kind of society/charity used in the experiment. The conclusions drawn seem to be very broad, which is why it becomes even more important to consider whether the type of sample used in the experiment is relevant for the kinds of charities the results are alluding to.

We acknowledge that our study examines a specific population for studying earmarking effects. However, importantly, studies on earmarking do not typically focus on a “typical donor” population either. Recent work has used general-population samples recruited via online platforms such as Prolific and MTurk (Esterzon et al., 2022; Özer et al., 2024) or nationally representative samples from 25 countries (Fuchs et al., 2020). These studies consistently find a positive effect of earmarking across varied demographic and cultural contexts, suggesting that the underlying mechanism is not specific to a particular donor group.

Moreover, while we acknowledge that academics may not be a primary target audience for large-scale fundraising campaigns, there is no theoretical or empirical reason to believe they differ systematically from the general population on dimensions relevant to prosocial behavior. In fact, evidence points in the opposite direction: higher education was found to be associated with increased charitable giving (Nakamura et al., 2025), and both empirical studies (Shaker & Palmer, 2012) and reports from institutional campaigns from U.S. universities (e.g., Anthony, 2023; WCU, 2023; UTSA, 2025) show that faculty and staff frequently engage in internal giving campaigns. These donations often support causes aligned with science, education, and equity—domains closely related to the professional identity of academic psychologists.

Most importantly, however, the primary focus of our study is not on increasing charitable donations, but on testing an intervention—earmarking—aimed at increasing survey response rates. As noted previously, we believe that academics represent a particularly difficult population to motivate for survey participation, and are therefore suited for investigating the research question at hand.

Nevertheless, we acknowledge that our sample has distinctive features that may limit the generalizability of our findings—most notably participants’ extensive experience with experimental manipulations, which can dampen the impact of such manipulations (Chandler et al., 2015; Krefeld-Schwalb et al., 2024). We now acknowledge this limitation more explicitly in the revised limitations section and call for future research to examine whether less experienced or more diverse populations respond differently, providing more generalizable insights into earmarking effects.

• The choice of the charity makes sense given the target audience, but it is unclear whether it is one that participants would care deeply about especially in comparison to charities that are usually studied in this literature. Did the authors perhaps ask questions to gauge whether participants trusted the Society to choose the right causes, spend the money fully et cetera? Similarly, was this a charity where participants likely cared enough about it in general to care about how their donation money would be spent?

Thank you for raising these important points. We acknowledge that our survey did not include explicit measures of participants’ trust in SIPS or the extent to which they valued its mission. While we believe many participants likely had strong confidence in SIPS’s stewardship—given its sustained engagement in the academic psychology community, its strong commitment to open and transparent research practices, and annual conferences that attract 500–1,000 attendees—we recognize that individual perceptions of trust still might vary. Likewise, concerning mission importance, we deliberately chose SIPS, a charity deeply embedded in participants’ professional community, to leverage identity congruence and value alignment, factors that have been shown to increase donation willingness (Chapman et al., 2025; Kesberg & Keller, 2021), however, we cannot be sure that mission importance was high for every participant. We now explicitly highlight this limitation in the revised manuscript and recommend that future research include direct measures of trust and mission importance, to clarify their moderating role in earmarking effects.

• One of my main concerns when reading about the treatment conditions was whether the purposes were sufficiently different for participants to care enough to want to complete the study and be able to earmark. The authors briefly touch upon this in the conclusion, but this may be a more significant factor than presented because, in the real world, charities, especially international ones, are donating to very disparate causes with a lot of differentiation. For instance, when donating to something like UNICEF, you can often choose between different causes and different countries and, therefore, it is much more likely that those who are donating will care about which cause their donation goes to. In comparison, roughly the same set of recipients between three purposes that are all to do with helping academics within the field of psychology is unlikely to evoke the same reaction or interest in differentiating between the various causes. It may also, in general, not be a set of causes that participants have strong feelings about in any case, again especially in comparison with the types of recipients that large charities, whether national or international, help.

We appreciate the thoughtful observation that the three SIPS purposes offered by the organization—supporting an open-access preprint platform and two forms of travel grants—are much more closely related than the highly differentiated options donors typically encounter at large humanitarian NGOs. We fully agree that this limited cause differentiation is a genuine limitation of our design when it comes to assessing the external validity of our findings. At the same time, that very homogeneity provides a useful feature: because the choices are not very heterogeneous, any increase we observe can likely only stem from the perceived impact pathway of earmarking, and not from strong pre-existing cause preferences. In other words, our study offers a conservative test of the mechanism. If earmarking produces a measurable uplift even when the available options are considerably similar, it should work at least as well—and plausibly better—when donors can choose among more differentiated causes. To make this rationale explicit, we have expanded the Discussion to describe our design as a conservative test and to call for future research that systematically varies cause differentiation as a potential moderator of the earmarking effect.

• In the analysis, it is unclear why "consent" is a dependent variable of interest. Participants who consented may or may not have reached the treatment page so it’s not obvious what information can be gleaned from this, and therefore it’s a bit odd that this is a main dependent variable in the analysis.

Thank you for highlighting the point with the “Consent” variable. We noticed that we provoked this issue in how we labeled and described this outcome variable. We recognize that this label may have been misleading, and we appreciate the opportunity to clarify both the variable’s role and how it fits within our experimental design.

In our study, treatment exposure occurred in the invitation email, which participants received before deciding whether to click through to the survey. As such, participants who chose not to enter the survey had already been exposed to the treatment condition. Therefore, the variable we initially called “consent” does not represent pre-treatment baseline behavior or a decision uninfluenced by condition. What we were capturing with that variable was whether participants clicked on the survey link and began the survey (indicated consent on the first page of the study). As suggested by the editor, a more accurate and informative label for this outcome is “survey begun.” We have revised the manuscript accordingly, updating all references to the variable across the text, tables, and figures to reflect this clarification.

• The authors summarize their power calculation, but it would be helpful to know whether a change of 2 percentage points would be meaningful in this context, as it seems rather low. It would also be helpful to know what the basis is for the 10% assumed participation rate, unless that is simply an example in which case stating that would add

---

## [Decision Letter · Decision Letter 1]

9 Jul 2025

PONE-D-25-07209R1Earmarking donations to boost study participation? Evidence from a field experimentPLOS ONE

Dear Dr. Raff,

Thank you for submitting your manuscript to PLOS ONE. We have reached a decision of "minor revision" (without external review). R1 is satisfied with the revisions as they address their points raised. R2 recommends a minor revision, asking you to do a better job in interpreting the substantive effects of the treatment and to discuss the wider implications of whether earmarking is overall a good strategy to boost donations, especially given a host of literature on the negative performance effects of earmarking.

We agree with R2's points, even though we are also aware that some of these asks fall out of the scope of your analysis. Please do a good faith effort to address these points. We would recommend you take a look at these studies on the effectiveness of earmarking, which should help contextualize the findings.

https://doi.org/10.1086/736339

https://doi.org/10.1111/rego.12632

https://doi.org/10.1017/S0020818323000085

Please note: the revised version will not be sent back to the reviewers. We hope this will accelerate the decision-making process.

We look forward to receiving your revised manuscript.

Kind regards,

Bernhard Reinsberg, Ph.D

Academic Editor

PLOS ONE

Journal Requirements:

Reviewers' comments:

Reviewer's Responses to Questions

**Comments to the Author**

1. If the authors have adequately addressed your comments raised in a previous round of review and you feel that this manuscript is now acceptable for publication, you may indicate that here to bypass the “Comments to the Author” section, enter your conflict of interest statement in the “Confidential to Editor” section, and submit your "Accept" recommendation.

Reviewer #2: All comments have been addressed

Reviewer #3: (No Response)

2. Is the manuscript technically sound, and do the data support the conclusions?

Reviewer #2: Partly

Reviewer #3: Yes

3. Has the statistical analysis been performed appropriately and rigorously? 

Reviewer #2: Yes

Reviewer #3: I Don't Know

4. Have the authors made all data underlying the findings in their manuscript fully available?

Reviewer #2: Yes

Reviewer #3: Yes

5. Is the manuscript presented in an intelligible fashion and written in standard English?

Reviewer #2: Yes

Reviewer #3: Yes

6. Review Comments to the Author

Reviewer #2: After careful consideration i believe the authors have adequately addressed my comments and concerns.

Reviewer #3: This article presents findings from a survey experiment on whether allowing participants to earmark charitable contributions given as an incentive for participation increases the response rate. Counter to expectations derived from literature on charitable giving, it finds that it does not. The article is clearly written and makes a succinct point. I think it is worthy of publication. However, I would suggest some further revisions to clarify the findings as well as their contribution and implications.

Most importantly, while the brief literature review covers key explanations for why earmarking should encourage charitable giving, it does so in a theoretical register. To help interpret the findings, the authors should provide substantive discussion of the findings of this literature, including the type of effects and their magnitude. For example, the article states that earmarking encourages charitable giving. However, it would be helpful to clarify whether it incentivises people to give who otherwise would not, or if it increases the amount they give. Relatedly, the authors should provide evidence on the magnitude of the effect for charitable giving found in previous studies to provide a sense of what effects we might be looking for in this experiment. Importantly, there must be some minimum level of effectiveness of earmarking for the research to be relevant, so that should be clearly stated. (This is the most important mechanism to provide substantive insights on, but it would be helpful to also provide brief substantive discussion of the other mechanisms highlighted in the literature review).

As a second point, the article claims that there is no evidence of harms of earmarking, and it might yield benefits. I was sceptical of this claim for two reasons: (1) As I understand it, earmarking is bad for charities, as it prevents them from using funds in the most effective way; and (2) presumably there is some additional cost to the researchers, even if marginal, to manage this data. The claims here should be clarified accordingly.

Third, I wondered if the authors might expand a bit more on what they think explains the lack of results. This may be easier to clarify once the expectations are more clearly stated (by drawing out the substantive content in the literature review as noted in point 1)

Minor points:

- The response rates of this study and other studies of 9-10% are first mentioned in the analysis strategy. I wondered if this could be mentioned earlier to further motivate the study.

- The first sentence of the abstract: “Charitable donations are often the best way to incentivise study participation” – requires citation and/or evidence, or should be toned down.

- It would be helpful to clarify throughout that this is about increasing response rates to surveys.

- The article states that “affluent or time-poor individuals may face opportunity costs that no realistic cash payment within the study budget can offset [15, 16]. Under such circumstances, even modest personal incentives are unlikely to attract these participants.” � but then presumably donation-based incentives also wouldn’t impact them? If there is evidence that people would be more persuaded by a donation than direct payment, stating it more explicitly would be important. Otherwise, I think the justification can rest on the idea that sometimes it is not appropriate to pay respondents and that it can be difficult (especially with data protection/management requirements, and for online surveys where you would not be able to hand cash directly to respondents)

- The article mentions the importance of trust to earmarking – but I wondered if it is also possible that if people really trust the organisation, they don’t care about earmarking because they figure the organisation can decide what to do with the resources more effectively than they can.

- On the ethics statement, clarify the language - The study was not anonymous, rather no identifying data was collected/respondents remained anonymous

- I agree with one of the previous reviewers’ comments that a brief reflection on the possibility that the $5 donation had no effect on participation at all is worth mentioning – even though of course the study did not test this, pointing it out as a pathway for future study may be valuable.

7. PLOS authors have the option to publish the peer review history of their article (what does this mean? ). If published, this will include your full peer review and any attached files.

**Do you want your identity to be public for this peer review?** For information about this choice, including consent withdrawal, please see our Privacy Policy .

Reviewer #2: **Yes: ** Stefanos Balaskas

Reviewer #3: No

---

## [Author Response · Author response to Decision Letter 2]

14 Aug 2025

Response to the Editor

Thank you for submitting your manuscript to PLOS ONE. We have reached a decision of "minor revision" (without external review). R1 is satisfied with the revisions as they address their points raised. R2 recommends a minor revision, asking you to do a better job in interpreting the substantive effects of the treatment and to discuss the wider implications of whether earmarking is overall a good strategy to boost donations, especially given a host of literature on the negative performance effects of earmarking.

Thank you for the positive feedback and for giving us the opportunity to revise the manuscript and address the remaining points raised by Reviewer 3. Guided by your comments and those of the reviewer, we have undertaken a further revision. Below we respond point-by-point to the specific issues that have been raised (original comments are italicized).

We agree with R2's points, even though we are also aware that some of these asks fall out of the scope of your analysis. Please do a good faith effort to address these points. We would recommend you take a look at these studies on the effectiveness of earmarking, which should help contextualize the findings.

https://doi.org/10.1086/736339

https://doi.org/10.1111/rego.12632

https://doi.org/10.1017/S0020818323000085

Please note: the revised version will not be sent back to the reviewers. We hope this will accelerate the decision-making process.

Thank you for providing these references. Combined with Reviewer 3’s request to consider the potential harms of earmarking, this prompted us to examine the issue more closely and to qualify our recommendation regarding its use. In the previous version, we had already acknowledged that earmarking may be suboptimal for charities because it reduces their flexibility in allocating funds. We now also highlight evidence from Heinzel et al. (2023) and Heinzel et al. (2025a, b), showing that earmarking can be associated with worse project performance in the context of international development organizations. Accordingly, in the implications section we now state that earmarking should only be implemented when its potential advantages clearly outweigh these downsides.

Response to Reviewer #3

This article presents findings from a survey experiment on whether allowing participants to earmark charitable contributions given as an incentive for participation increases the response rate. Counter to expectations derived from literature on charitable giving, it finds that it does not. The article is clearly written and makes a succinct point. I think it is worthy of publication. However, I would suggest some further revisions to clarify the findings as well as their contribution and implications.

Most importantly, while the brief literature review covers key explanations for why earmarking should encourage charitable giving, it does so in a theoretical register. To help interpret the findings, the authors should provide substantive discussion of the findings of this literature, including the type of effects and their magnitude. For example, the article states that earmarking encourages charitable giving. However, it would be helpful to clarify whether it incentivises people to give who otherwise would not, or if it increases the amount they give. Relatedly, the authors should provide evidence on the magnitude of the effect for charitable giving found in previous studies to provide a sense of what effects we might be looking for in this experiment. Importantly, there must be some minimum level of effectiveness of earmarking for the research to be relevant, so that should be clearly stated. (This is the most important mechanism to provide substantive insights on, but it would be helpful to also provide brief substantive discussion of the other mechanisms highlighted in the literature review).

We appreciate your positive feedback on our manuscript. We also thank you for highlighting the importance of specifying the nature and magnitude of earmarking effects in prior research. We have revised the manuscript to address these points more explicitly:

First, we have now more clearly specified the nature of earmarking effects in charitable giving. In the revised manuscript, we now explicitly specify that, while theoretically earmarking could increase both the willingness to donate and the amount donated, empirical findings more consistently show effects on willingness to donate. We argue that it is precisely this increased willingness to donate rather than increases in donation amounts that positions earmarking as a potentially valuable intervention in the context of incentivizing survey participation.

Second, regarding your suggestion to discuss the magnitude of earmarking effects from prior studies, we agree that setting clear expectations is important. However, we are cautious about making direct comparisons because previous research has examined different outcomes (e.g., webpage engagement in Costello & Malkoc, 2022; willingness to donate in Fuchs et al., 2020). Since our study is the first to test the effect of earmarking on study participation, we considered it more appropriate to base expectations on effect sizes from studies that directly target this outcome—thereby avoiding an apples-to-oranges comparison between willingness to donate and willingness to participate in a study.

In this regard, meta-analyses of web- and electronic-based survey research report that study participation can increase substantially—by odds ratios ranging from 1.39 to 2.43—when financial incentives are offered (David & Ware, 2014; van Gelder et al., 2018; Edwards et al., 2023). Assuming a baseline participation rate of 10%, these odds ratios translate into increases of roughly 4 to 12 percentage points, illustrating that participation can respond substantially to changes in incentive structures. Based on this, we set a 2 percentage-point increase (from a 10% baseline) as the minimum effect size that would render earmarking practically worthwhile in our context. From the researcher’s perspective, an effect of this magnitude could potentially justify the additional organizational effort required to implement earmarked donations, including establishing a collaboration with a charity, ensuring proper fund allocation, and managing follow-up communication. From the charity’s perspective, the expected increase in donations could potentially outweigh the downsides of earmarking. This rationale is now explicitly reflected in the revised manuscript.

As a second point, the article claims that there is no evidence of harms of earmarking, and it might yield benefits. I was sceptical of this claim for two reasons: (1) As I understand it, earmarking is bad for charities, as it prevents them from using funds in the most effective way; and (2) presumably there is some additional cost to the researchers, even if marginal, to manage this data. The claims here should be clarified accordingly

We appreciate this point and agree that the potential downsides of earmarking—such as reduced flexibility for charities and additional administrative burdens for researchers—are important to acknowledge. Our original statement that earmarking “does not harm” referred specifically to participation rates. We have now made this explicit and qualified our recommendation to use earmarking by noting that its implementation should always be weighed against potential downsides identified in prior research (e.g., reduced flexibility, worse project performance) and the added administrative costs for researchers, and should therefore be implemented only when its potential advantages clearly outweigh these downsides.

Third, I wondered if the authors might expand a bit more on what they think explains the lack of results. This may be easier to clarify once the expectations are more clearly stated (by drawing out the substantive content in the literature review as noted in point 1)

As we now make clear, thanks to your earlier suggestion, our expectation was that earmarking would motivate more people to participate. One possible reason it did not do so in this context is a difference in psychological ownership of the donated funds between donations made with one’s own money and those made through externally funded donation incentives. While we had implied this previously, we now make it more explicit in the future research section:

“A promising direction for future studies is to examine the role of psychological ownership in differentiating donations made with one’s own money from externally funded donation incentives, as theorized before, which could be a reason why earmarking in this particular instance did not prove effective.”

Minor points:

- The response rates of this study and other studies of 9-10% are first mentioned in the analysis strategy. I wondered if this could be mentioned earlier to further motivate the study.

We appreciate the suggestion. In our view, the reference to response rates from other studies serves primarily to provide a plausible baseline for the sensitivity analysis, rather than to motivate the study itself. For this reason, we believe it fits most naturally within the analysis strategy section. If you or the editor still believe this information should appear earlier, we would be happy to try to integrate it accordingly.

- The first sentence of the abstract: “Charitable donations are often the best way to incentivise study participation” – requires citation and/or evidence, or should be toned down.

We thank you for this comment and have revised the first sentence of the abstract to state more cautiously that, in many cases, charitable donations can be the most suitable form of incentivization. This is supported by reasoning presented later in the manuscript, where we note that budgetary, ethical, or participant-specific constraints may at times preclude the use of personal incentives.

- It would be helpful to clarify throughout that this is about increasing response rates to surveys.

Following your suggestions, we have revisited the manuscript to ensure that our focus on survey response rates is consistently clear. While the abstract and introduction already made this focus explicit, we have further reinforced it by adding the phrase “in attracting respondents for surveys” when describing prior research and by specifying “to increase response rates” when discussing the importance of understanding how to implement donation incentives optimally. We hope that these adjustments make the objective of the paper clearer from the outset.

- The article states that “affluent or time-poor individuals may face opportunity costs that no realistic cash payment within the study budget can offset [15, 16]. Under such circumstances, even modest personal incentives are unlikely to attract these participants.” � but then presumably donation-based incentives also wouldn’t impact them? If there is evidence that people would be more persuaded by a donation than direct payment, stating it more explicitly would be important. Otherwise, I think the justification can rest on the idea that sometimes it is not appropriate to pay respondents and that it can be difficult (especially with data protection/management requirements, and for online surveys where you would not be able to hand cash directly to respondents)

Thank you for this comment and for prompting this clarification. Indeed, the findings by Khan et al. (2020) show that when the amount that can be paid per person is small, donation incentives can be more motivating than direct payments. While both incentive types may have limited cognitive valuation due to the low amount, donations add an affective component that can make them more compelling. We now make this mechanism explicit in the manuscript.

- The article mentions the importance of trust to earmarking – but I wondered if it is also possible that if people really trust the organisation, they don’t care about earmarking because they figure the organisation can decide what to do with the resources more effectively than they can.

Thank you for this thoughtful observation. We already note in the manuscript that trust is likely of lower importance in the context of donation incentives for study participation, given the relatively small amounts involved and the lower psychological ownership of the donated funds. This suggests that the trust-based mechanism through which earmarking can increase donations may be weaker in this setting. As you point out, it is also possible that when trust in the organization is high, offering earmarking may not increase donations via this pathway because participants believe the organization can allocate resources more effectively than they can. However, even when trust does not play a role in a particular instance, this does not necessarily mean that earmarking would have no positive effect. Arguably, the main reason for its motivational potential lies not in increasing trust, but in its ability to satisfy the basic psychological needs for autonomy and competence, as explained by cognitive evaluation theory.

- On the ethics statement, clarify the language - The study was not anonymous, rather no identifying data was collected/respondents remained anonymous

Thank you for pointing this out. We have removed the wording that described the study as “anonymous,” as the manuscript already specifies that no identifying data were collected.

- I agree with one of the previous reviewers’ comments that a brief reflection on the possibility that the $5 donation had no effect on participation at all is worth mentioning – even though of course the study did not test this, pointing it out as a pathway for future study may be valuable.

Again thank you for this comment. It prompted us to adapt the first limitation referring to the low amount. While we already note that the $5 incentive may have been too small, we now also connect this with the specific characteristics of donation incentives identified earlier, namely, lower psychological ownership of the funds. It may be the case that, due to this lower psychological ownership, earmarking within a donation incentive only has a measurable effect at higher amounts (compared to the charitable donation scenario).

References:

Costello, J. P., & Malkoc, S. A. (2022). Why Are Donors More Generous with Time Than Money? The Role of Perceived Control over Donations on Charitable Giving. The Journal of Consumer Research, 49(4), 678–696. https://doi.org/10.1093/jcr/ucac011

David, M. C., & Ware, R. S. (2014). Meta-analysis of randomized controlled trials supports the use of incentives for inducing response to electronic health surveys. Journal of Clinical Epidemiology, 67(11), 1210–1221. https://doi.org/10.1016/j.jclinepi.2014.08.001

Edwards, P. J., Roberts, I., Clarke, M. J., DiGuiseppi, C., Woolf, B., & Perkins, C. (2023). Methods to increase response to postal and electronic questionnaires. Cochrane Database of Systematic Reviews, 2023(11), MR000008.

https://doi.org/10.1002/14651858.MR000008.pub5

Fuchs, C., de Jong, M. G., & Schreier, M. (2020). Earmarking Donations to Charity: Cross-cultural Evidence on Its Appeal to Donors Across 25 Countries. Management Science, 66(10), 4820–4842. https://doi.org/10.1287/mnsc.2019.3397

Heinzel, M., Cormier, B., & Reinsberg, B. (2023). Earmarked Funding and the Control–Performance Trade-Off in International Development Organizations. International Organization, 77(2), 475–495. https://doi.org/10.1017/S0020818323000085

Heinzel, M., Reinsberg, B., & Zaccaria, G. (2025a). Core funding and the performance of international organizations: Evidence from UNDP projects. Regulation & Governance, 19(3), 957–976. https://doi.org/10.1111/rego.12632

Heinzel, M., Reinsberg, B., & Siauwijaya, C. (2025b). Understanding Resourcing Trade-offs in International Organizations: Evidence from an Elite Survey Experiment. The Journal of Politics. https://doi.org/10.1086/736339

Khan, U., Goldsmith, K., & Dhar, R. (2020). When Does Altruism Trump Self-Interest? The Moderating Role of Affect in Extrinsic Incentives. Journal of the Association for Consumer Research, 5(1), 44–55. https://doi.org/10.1086/706512

van Geld

---

## [Editor Report · Decision Letter 2]

18 Aug 2025

Earmarking donations to boost study participation? Evidence from a field experiment

PONE-D-25-07209R2

Dear Dr. Raff,

We’re pleased to inform you that your manuscript has been judged scientifically suitable for publication and will be formally accepted for publication once it meets all outstanding technical requirements. Congratulations to a fine piece of research!

Once again, congratulations on your fine contribution, and thank you for publishing with PLOS ONE.

Kind regards,

Bernhard Reinsberg, Ph.D

Academic Editor

PLOS ONE

Additional Editor Comments (optional):

All comments addressed.